# The First Genome-Wide Mildew Locus O Genes Characterization in the *Lamiaceae* Plant Family

**DOI:** 10.3390/ijms241713627

**Published:** 2023-09-04

**Authors:** Andolfo Giuseppe, Ercolano Maria Raffaella

**Affiliations:** Department of Agricultural Sciences, University of Naples “Federico II”, Via Università 100, Portici, 80055 Naples, Italy

**Keywords:** medicinal plants, MLO, powdery mildew, functional inference, disease resistance

## Abstract

Powdery mildew (PM) is a widespread plant disease that causes significant economic losses in thousands crops of temperate climates, including *Lamiaceae* species. Multiple scientific studies describe a peculiar form of PM-resistance associated at the inactivation of specific members of the Mildew Locus O (MLO) gene family, referred to as mlo-resistance. The characterization of *Lamiaceae* MLO genes, at the genomic level, would be a first step toward their potential use in breeding programs. We carried out a genome-wide characterization of the MLO gene family in 11 *Lamiaceae* species, providing a manual curated catalog of 324 MLO proteins. Evolutionary history and phylogenetic relationships were studied through maximum likelihood analysis and motif patter reconstruction. Our approach highlighted seven different clades diversified starting from an ancestral MLO domain pattern organized in 18 highly conserved motifs. In addition, 74 *Lamiaceae* putative PM susceptibility genes, clustering in clade V, were identified. Finally, we performed a codon-based evolutionary analysis, revealing a general high level of purifying selection in the eleven *Lamiaceae* MLO gene families, and the occurrence of few regions under diversifying selection in candidate susceptibility factors. The results of this work may help to address further biological questions concerning MLOs involved in PM susceptibility. In follow-up studies, it could be investigated whether the silencing or loss-of-function mutations in one or more of these candidate genes may lead to PM resistance.

## 1. Introduction

The mint family (*Lamiaceae*) is an important group of medicinal plants. It consists of more than 7000 aromatic species, distributed nearly worldwide, and commonly widespread in regions of lands around the Mediterranean area [1,2]. Some species (such as mint, basil, lavender and rosemary) are famous for their use in essential oil (EO) production worldwide. EOs can be extracted from several parts of medicinal plants [3] for foods, cosmetic, cleaning, and pharmaceutical industries applications [4,5]. Generally, EO yield is correlated with the number and distribution of glandular trichomes, in which oil components are synthesized and stored [6]. *Lamiaceae* pathogens can certainly compromise plant growth, directly or indirectly affecting the quality and abundance of EOs. Among them, the powdery mildew (PM) is considered to be one of the main biotic threats for the cultivation of species of the *Lamiaceae* species [7,8]. PM colonizes young stems and leaves producing the characteristic white mycelium; when the attack is very severe, stems can be completely killed. The use of resistant cultivars can significantly reduce the amount of fungicides necessary to control PM.

Several studies revealed that the loss of function of Mildew Locus O (MLO)-specific gene members confers resistance to powdery mildew [9,10,11,12,13]. The numbers of members in the MLO family in the plant kingdom ranged from three in *Volvox carteri* to 55 in *Triticum aestivum* [14]. Previous inference analyses grouped MLO proteins in distinct subfamilies, or phylogenetic clades, with a varying number of clades depending on the analyzed species [14,15,16]. Available scientific literature indicates that two clades (IV and V in monocots and dicots, respectively) host MLO proteins associated with PM susceptibility [14,15,17,18]. Indeed, the inactivation of these genes by gene silencing, genome editing, or TILLING leads to a peculiar form of resistance (known as mlo-resistance) based on the enhancement of pre-penetrative defense responses [19,20,21]. The broad-spectrum effectiveness and durability makes mlo-resistance a general breeding strategy against PM of cultivated species.

Although MLO genes have been studied in several species [15], they have not yet been identified in *Lamiaceae* plant family. The growing interest for medicinal properties as well as for economic importance in *Lamiaceae* species has led to the sequencing of several genomes in recent years [22,23,24,25,26,27,28,29], thus providing an opportunity to explore the variability of their MLO gene families.

In this study, we used the genome sequences of *Callicarpa americana*, *Mentha longifolia*, *Ocimum basilicum*, *Origanum majorana*, *Origanum vulgare*, *Rosmarinus officinalis*, *Salvia bowleyana*, *Salvia hispanica*, *Salvia miltiorrhiza*, *Salvia officinalis* and *Salvia splendens* to annotate the corresponding MLO gene families. Through a phylogenetic analysis and the characterization of MLO motifs, we gained information with respect to their structure, evolutionary history and function. Furthermore, genetic diversity studies were performed to improve knowledge on the main MLO selection force driving evolution associated with PM susceptibility. A better understanding of *Lamiaceae* MLOs will promote progress in both basic research and practical breeding activities aimed to select PM-resistant cultivars in these species.

## 2. Results

### 2.1. MLO Gene Annotation in Eleven Lamiaceae Species

In order to identify the genes encoding for the MLO proteins in *Lamiaceae* plant family, a protein domain search analysis into the *C. americana*, *M. longifolia*, *O. basilicum*, *O. majorana*, *O. vulgare*, *R. officinalis*, *S. bowleyana*, *S. hispanica*, *S. miltiorrhiza*, *S. officinalis* and *S. splendens* proteomes was performed. Since the official gene annotation of *M. longifolia* and *Salvia officinalis* genome assemblies are not available (Appendix A), the loci harboring a typical MLO domain were identified by using an in-house-built pipeline described in the Materials and Methods section.

A total of 324 MLO genes were identified and finely characterized into the 11 *Lamiaceae* genomes (Table 1; Appendix A). MLO gene family size in *Lamiaceae* ranged from 20 to 54 members in *O. vulgare* and *S. splendens*, respectively (Table 1). A conspicuous diversification in the size of the MLO families across species was revealed, while within the same genus (e.g., *Origanum* sp., *Salvia* sp.) narrow differences were observed (Table 1). A positive relationship between the size of a predicted gene set in each genome assembly and the relative number of MLO genes was displayed by correlation analysis (Pearson’s r = 0.769).

The size of manually curated MLO genes identified in *Lamiaceae* genomes, composed of 11 to 14 exons, ranged between 3591 bp (*O. majorana*) and 5105 bp (*S. bowleyana*) (Table 2 and Appendix A), while the number of amino acid residues encoded by these MLO genes ranged from 407 to 520 (Table 2). The average length of MLO domain (pfam03094), archived in “The Protein Family” (Pfam) database, resulted of 367 amino acids (Appendix A). The MLO genes predicted and characterized in the *M. longifolia* and *S. officinalis* genomes [23,29] were consistently similar to the ones annotated in remaining analyzed species in terms of structural genomic properties (Table 2). The longest MLO gene model belonged to *S. splendens* (KAG6400176.1), with a total length of 15,015 nucleotides (Appendix A).

### 2.2. Identification of MLO Genes Associated to PM Susceptibility

The MLO-Pfam domain sequences of *C. americana*, *M. longifolia*, *O. basilicum*, *O. majorana*, *O. vulgare*, *R. officinalis*, *S. bowleyana*, *S. hispanica*, *S. miltiorrhiza*, and *S. splendens* MLO proteins were used to infer phylogenetic distances among them and with respect to functionally validated MLOs in other plant species (Appendix A). In total, 293 MLO proteins collapsed in seven robust phylogenetic clades (bootstrap index ≥ 95) (Figure 1; Appendix A). These were designated with the Roman numerals from I to VII, based on the position of Arabidopsis and monocot MLO homologs reported in the previous studies [14,33].

Clade I includes fifty-one MLO proteins, three of which are annotated in Arabidopsis (AtMLO4, AtMLO11 and AtMLO14), 10 in *S. splendens,* five in *M. longifolia*, five *O. basilicum*, five in *S. hispanica*, four in *S. bowleyana*, four in *C. americana*, four in *O. vulgare*, four in *R. officinalis,* four in *S. miltiorrhiza*, and three in *O. majorana*. Clade II groups forty-eight MLO homologs, three of which identified in Arabidopsis (AtMLO1, AtMLO13 and AtMLO15), six in *R. officinalis* and *O. basilicum*, five in *M. longifolia*, and four in all remaining analyzed species (*C. americana*, *O. majorana*, *O. vulgare*, *S. bowleyana*, *S. hispanica*, *S. miltiorrhiza* and *S. splendens*).

A common ancestor (bootstrap support 98/100) in the evolutionary history of clades III, IV, V, VI and VII was found (Figure 1). Clade III includes thirty-six *Lamiaceae* homologs (eight in *S. splendens*; seven in *O. basilicum*; five in *S. hispanica*; three in *C. americana* and *R. officinalis*; and two in *M. longifolia*, *S. bowleyana*, *S. miltiorrhiza, O. majorana* and *O. vulgare*) together with five Arabidopsis proteins (AtMLO5, AtMLO7, AtMLO8, AtMLO9 and AtMLO10). Phylogenetic data also provided evidence for a common ancestor originating clades III and IV (Appendix A). Clade IV, which is conventionally associated with MLO proteins (TaMLO_A1b, TaMLO_B1, HvMLO1 and OsMLO3) from monocot species, also contains 13 *Lamiaceae* MLO proteins (two in *S. splendens*, *O. basilicum* and *M. longifolia*; one in *C. americana*, *O. majorana, O. vulgare, R. officinalis, S. bowleyana*, *S. hispanica* and *S. miltiorrhiza*).

Nine *M. longifolia*, *O. basilicum* and *S. splendens*, seven *S. hispanica* and *S. miltiorrhiza*, six *S. bowleyana* and *O. vulgare*, five *C. americana* and *R. officinalis*, and four *O. majorana* proteins cluster together in the phylogenetic clade V, including all the dicot MLO homologs experimentally shown to act as PM susceptibility factors. Within the clade V, based on a bootstrap index ≥ 53, two subclades (Va and Vb) were distinguishable, including eight functionally characterized MLO proteins (AtMLO2, AtMLO6, AtMLO12, SlMLO1, PsMLO1, CaMLO2, LjMLO1 and MtMLO1) in the subclade Va and one (VvMLO3) in the subclade Vb (Figure 1). Clade VI is located on an ancestral position (Appendix A) and included AtMLO3 protein and 25 *Lamiaceae* MLOs. In particular, four MLO proteins annotated in *O. basilicum* and three in *R. officinalis*, *S. splendens* and *S. hispanica*, while only two AtMLO3 homologs were found in the remaining analyzed species (Figure 1; Appendix A). Finally, clade VII includes 13 *Lamiaceae* MLO proteins (three in *S. splendens*; two *R. officinalis* and *S. bowleyana*; and one in *C. americana*, *M. longifolia*, *O. basilicum*, *O. vulgare*, *S. hispanica* and *S. miltiorrhiza*).

### 2.3. Characterization of Clade-Specific Amino Acid Motifs

To explain the evolutionary routes emerging in the phylogenetic tree of *Lamiaceae* MLO gene family, a motif-based sequence analysis was carried out. Towards this goal, the MLO Pfam domain (pfam03094) of our protein dataset was divided into 30 ungapped motifs by MEME (Appendix A), which were used by MAST for a sequence pattern analysis. The motifs structure of MLO Pfam domains for each collapsed clade was showed in Figure 2.

MLO Pfam domain was outlined from a total of 43 motifs since some MEME-motifs were detected multiple times along the protein sequences (Figure 2). The number of MLO-related motifs increased from clade VII (27) to clades III and V (36). Our analysis has underlined an ancestral MLO-core structure including 23 motifs highly conserved between the seven clades, of which 78% (18 out of 23) were organized in four blocks (red rectangles in Figure 2). Starting from this ancestral structure, the MLO family has diversified and functionally specialized in the seven phylogenetic clades. The presence/absence of 20 additional motifs contribute to generate the seven clade-specific patterns (Appendix A).

Clade V MLO homologs showed the most complex profile, beside the red conserved blocks, six clade-specific motifs were identified between the red blocks three and four (Figure 2).

An additional block composed from four motifs (MEME-8, MEME-29, MEME-14 and MEME-16) was present only in clades III, IV, V and VI (blue rectangles in Figure 2). Noteworthily, three motifs (MEME-26, MEME-17 and MEME-25) were simultaneously conserved between clade IV and clade V (red lines in Figure 2). Analyzing in more detail the MLO domain structure of all the dicot MLO homologs experimentally tested so far for PM susceptibility (AtMLO2, AtMLO6, AtMLO12, SlMLO1, PsMLO1, CaMLO2, LjMLO1, MtMLO1 and VvMLO3), 30 conserved motifs were detected (Appendix A), suggesting that they are functionally related. Only 24% (18 out of 74) of *Lamiaceae* MLO proteins showed the conserved pattern of molecularly validated MLO genes for PM susceptibility (Appendix A). Diversely, the functionally well-characterized monocot MLOs (TaMLO_A1b, TaMLO_B1, HvMLO1 and OsMLO3) presented four conserved motifs that were absent in all *Lamiaceae* MLOs of clade IV.

### 2.4. Selection Pressure Acting on Lamiaceae MLO Phylogenetic Clades

In order to discover the direction and magnitude of natural selection acting on MLO protein coding genes in *C. americana*, *M. longifolia*, *O. basilicum*, *O. majorana*, *O. vulgare*, *R. officinalis*, *S. bowleyana*, *S. hispanica*, *S. miltiorrhiza*, *S. officinalis* and *S. splendens*, we used a combination of maximum likelihood and counting approaches implemented into the single-likelihood ancestor counting (SLAC) method [34].

Neutrality tests performed on the MLO gene family of each of the eleven *Lamiaceae* species, yielded π and ω values ranges from 0.342 to 0.431 and from 0.311 to 0.212, respectively (Table 3). On average, ~239 (max. 292; min. 180) negatively selected codon sites are statistically significant in each MLO gene family (Table 3). Comprehensive analyses reveal that negative selection has been acting against extreme polymorphic variants in *Lamiaceae* plant species. Indeed, only eleven positively selected codon sites were statistically supported in five out of eleven analyzed species (Table 3).

Similarly, a purifying selection (0.216–0.375 ω values) was observed into the seven phylogenetic clades (Table 3). Indeed, the average sequence identity ranges from 44.2% to 73% in clades V and VII, respectively. Likely, natural selection acts against mutations that have deleterious effects on functionally important amino acid residues. Single-codon analysis highlighted the presence of several negatively selected sites varying from 371 to 65 in the V and VII clades, respectively (Table 3). With respect to other phylogenetic clades, clade V displays a higher ω (0.375) value (Table 3), thus suggesting that it is subjected to a larger number of selective constraints. Indeed, 371 out of 954 negatively selected codon sites were also statistically significant (Table 3; Figure 3). Finally, a total of 407 positively selected codon sites (ω > 1) were identified along the MLO protein-coding-genes collapsed in clade V, of which only four statistically significant (Figure 3). The relaxed purifying selection might be the main driving force during the evolution of MLO genes associated with PM susceptibility.

## 3. Discussion

In the present study, we exploited available genomic resources to characterize the MLO gene family in cultivated *Lamiaceae*. A total of 324 MLO protein encoding genes from eleven medicinal plant genomes were identified. It was observed that a direct correlation between predicted gene set sizes and the number of MLO gene family members. For example, the MLOs annotated in the genome assemblies released by Bornowski et al. [25] were higher in genome with a greater gene set, and vice versa (Table 1). Diversely, the number of MLO genes of the *S. miltiorrhiza* (32,483 loci) [24] and *C. americana* (32,164 loci) [26] did not change significantly.

In addition, we also noted that, the tetraploid *O. basilicum* cultivar (Genovese) (2*n* = 4× = 48) and *S. splendens* (2*n* = 4× = 44) genomes (Bornowski et al., 2020; Jia et al., 2021), displayed half of MLO orthologs in multiple copies, consistent with their polyploidy [35]. These findings indicate that the whole-genome duplication events contributed to the expansion of MLO gene families [36]. All the MLO proteins identified in *Lamiaceae* genomes have amino acid lengths comparable to those of Arabidopsis AtMLO homologs, ranging from 460 to 593 residues [33]. Similarly, the de novo gene models predicted in *M. longifolia* and *S. officinalis* genomes, obtained in this work, showed structural features in line with the MLO gene annotation reported in other species (Table 2).

Relations of orthology between *Lamiaceae* MLO members were inferred based on phylogenetic relatedness [15,33,37]. Based on the functionally characterized MLO genes, the gene family is collapsed into six major clades, of which the clade IV was exclusively present in monocots, while the clade V was only present in dicots [33]. Iovieno et al. [14] reported for the first time the occurrence of MLOs belonging to dicotyledons in clade IV. Furthermore, some authors claim that the MLO genes are organized in seven clades [16], although not all plant species harbor members in all clades [38]. An extra clade (VIII) has been proposed, and various subclades have been identified by Iovieno et al. [14], but no consensus has been reached yet. Our maximum likelihood analysis assigned *Lamiaceae* MLO proteins to the seven evolutionary clades. We found considerable differences in the variation of the number of genes (from one to nine) per clade across species. Although Clade IV is considered exclusively composed of monocot MLO proteins [19], we found 14 *Lamiaceae* MLOs grouped in it, confirming recent findings [14]. In *Lamiaceae* clade IV, we distinguished a primary subgroup (bootstrap index = 99), including all the monocot homologs involved in the interaction with PM fungi (namely barley HvMLO, rice OsMLO3 and wheat TaMLO_A1 and TaMLO_B1) separated from the group including the 14 *Lamiaceae* MLOs. Probably, clade IV *Lamiaceae* MLO homologs have the same role of MLOs previously identified in [14].

Interestingly, 74 MLO homologs were found to cluster together in the phylogenetic clade V, including also all the dicot reference genes (namely Arabidopsis AtMLO2, AtMLO6 and AtMLO12; tomato SlMLO1; grapevine VvMLO3 and VvMLO4; tobacco NtMLO1; pepper CaMLO2 and barrel clover MtMLO1) functionally associated with PM susceptibility [17,37]. Similar to clade IV, it was determined that clade V is composed of two subclades, named Va and Vb (Figure 1). Subclade Va is composed of all the reference genes, except for VvMLO3, seven homologs belonging to *S. miltiorrhiza*, five to *C. americana*, seven to *S. officinalis*, nine to *M. longifolia*, four to *O. majorana*, six to *O. vulgare*, five to *R. officinalis*, six to *S. bowleyana*, seven to S. hispanica, nine for *S. splendens*, and nine for *O. basilicum*. The subclade Vb, includes VvMLO3, three members belonging to *S. splendens*, two to *R. officinalis* and *S. bowleyana*, and one to all the remaining *Lamiaceae* species. Our study provides information on 65 and 14 putative novel PM susceptibility factors in subclades Va and Vb, respectively. These MLO gene targets could be used for future breeding activities aimed to introduce mlo-resistance in cultivated species. Noteworthy, transcription activator-like effector nucleases (TALEN) and clustered regularly interspaced short palindromic repeat (CRISPR) technology can be used to inactivate PM susceptibility genes [20,21].

We reanalyzed the multiple alignment of MLO protein dataset to detect amino acid motifs that, being highly conserved, are predicted to play a major role for PM susceptibility function. For the first time, we found that each clade was characterized by a specific conserved motifs patter, implying that specific conserved profiles were likely required for subfamily-specific functions. In the *Lamiaceae* MLO gene family, the seven MLO clades were evolved from the ancient MLO domain structure. Subsequently, short duplication, insertion or deletion of specific motifs occurred in the MLO protein domain (Figure 2). Clade-related evolutionary patters exhibit some degree of local sequence conservation and diversification that could result in fold change or deterioration. Gene sequence divergence, recombination and duplications resulted the main driving forces for the gene diversification [39,40,41].

As expected, a very high conservation level (pairwise average identity: ~38%) was found in the MLO Pfam domain (Pfam03094) sequences, including the extracellular loops or transmembrane regions, previously found to be essential for MLO functionality and/or stability [42]. In addition, we detected a series of conserved motifs scattered in the clade V *Lamiaceae* MLO homologs. These might be specifically important for the evolution of functional isoforms associated with PM susceptibility. With this respect, our study complements the previous works on conserved amino acid residues and motifs [15,42,43], reporting the identification of a motif pattern specifically conserved in putative PM susceptibility factors.

In line with the identification of several conserved amino acidic motifs, the neutrality test addressed to infer evolutionary forces acting on *Lamiaceae* MLO homologs suggested a general high level of negative selection (Table 3). The negative selection contributes to low gene family variability, which was consistent with the hypothesis that highly conserved genes remain in the genome due to purifying selection. Most likely, the strong negative selection has likely contributed to the maintenance of consistent clade sizes across species. Interestingly, single-codon analysis of clade V *Lamiaceae* MLO homologs highlighted the occurrence of three protein regions that are likely under positive selection pressure (Figure 3), as reported by Ioieno et al. [15] for Cucurbitaceae and Rosaceae clade V MLO homologs. Generally, the gene copies with high variability in specific sites tend to be involved in plant–environment interactions, particularly biotic interactions [44]. Positive selection drives plant/pathogen co-evolution [45,46], in accordance with an “arms race” model [47,48,49]. Although the molecular interaction between MLO proteins and pathogen effectors is still elusive to date, it might be speculated that positively selected codon sites located in the intracellular C-terminus are implicated in pathogen sensing [15].

## 4. Conclusions

The comprehensive analysis of the MLO gene families in eleven economically important *Lamiaceae* species allowed us to assess the variability existing in this botanical family. Starting from the MLO core structure reconstructed in this work, we were able to depict the diversification that occurred in seven phylogenetic clades. For the first time, we highlighted the conserved motifs patter of each MLO clade. Our findings are important for the identification of a motif profile specifically associated with PM susceptibility. The conserved pattern of *Lamiaceae* MLOs in clade IV and V could be further investigated to identify putative PM susceptibility factors. Furthermore, the identification of codons under positive selection pressure in clade V members may direct the selection of appropriate candidate genes for further breeding activities aimed at introducing PM resistance.

## 5. Materials and Methods

### 5.1. Taxa Data Set

To investigate the evolution of MLO genes along the *Lamiaceae* plant family, we used the genomic data of eleven taxa (*Callicarpa americana*, *Mentha longifolia*, *Ocimum basilicum*, *Origanum majorana*, *Origanum vulgare*, *Rosmarinus officinalis*, *Salvia bowleyana*, *Salvia hispanica*, *Salvia miltiorrhiza*, *Salvia officinalis* and *Salvia splendens*). The genome and proteome sequences were retrieved from five (DRYAD, GigaDB, CNGBdb, GenBank, NGDC and GWH) public database (Appendix A). Moreover, for comparative purposes we also added 15 MLO genes identified in *Arabidopsis thaliana* retrieved from TAIRdb (www.arabidopsis.org, accessed on 1 June 2023) and 10 well-characterized reference MLO genes in other species (*A. thaliana*, *C. annuum*, *H. vulgare*, *L. japonicas*, *M. truncatula*, *O. sativa*, *P. sativum*, *S. lycopersicum*, *T. aestivum* and *V. vinifera*) (Appendix A).

### 5.2. Identification and Manual Curation of Eleven Lamiacaeae MLO Gene Families

In order to retrieve predicted *Lamiaceae* MLO genes, we scanned the *C. americana*, *L. angustifolia*, *O. basilicum*, *O. majorana*, *O. vulgare*, *R. officinalis*, *S. hispanica*, *M. longifolia*, *S. miltiorrhiza*, *S. officinalis* and *S. splendens* proteome data set (Appendix A) with the hidden Markov model of the MLO domain (Pfam: PF03094) using HMMER v3.0 with default parameters [50]. In addition, a local BLASTp search (e-value < 1 × 10^−5^) was performed by mapping A. thaliana amino acid sequences as queries (AtMLO1-AtMLO15 (Appendix A), NCBI accession numbers: NP_192169.1, NP_172598.1, NP_566879.1, NP_563882.1, NP_180923.1, NP_176350.1, NP_179335.3, NP_565416.1, NP_174980.3, NP_201398.1, NP_200187.1, NP_565902.1, NP_567697.1, NP_564257.1 and NP_181939.1) to the *Lamiaceae* protein data set. The protein domain architecture of HMMER and BLAST outputs was annotated using InterProScan [32] and conserved domain search [51] with default parameters.

The official gene annotation of *M. longifolia* and *S. officinalis* were not made available on GenBank (GCA001642375.2) and NGDC (GWHBJVP00000000), respectively. To identify the MLO coding sequences located on *M. longifolia* scaffolds and *S. officinalis* chromosomes, we performed a gene annotation using the Augustus (v2.5.5) pipeline [30] with Arabidopsis parameters. Only the predicted genes encoding proteins of more length, 50 amino acids and a confidence score > 0 were used to perform *M. longifolia* and *S. officinalis* MLO annotations.

### 5.3. Multiple Sequence Alignments and Phylogenetic Analysis

Sequence similarities were determined performing a ClustalW (v2.0) [52] multiple alignment with default settings, using the conserved MLO domain sequence of the PFAM database (ID: PF03094) as input. Phylogenetic analysis was performed by using newly identified *Lamiaceae* MLO homologs containing at least 50% of the full-length MLO domain. The dataset was completed with the whole Arabidopsis MLO protein family and the following proteins previously shown to act as PM susceptibility factors: pea PsMLO1, barley HvMLO, rice OsMLO2, pepper CaMLO2, tomato SlMLO1, grape VvMLO3, barrel clover MtMLO1, lotus LjMLO1, and wheat TaMLO1_A1b and TaMLO_B1 (Appendix A). All these sequences were extracted from the NCBI database (http://www.ncbi.nlm.nih.gov, accessed on 1 June 2023). Evolutionary relationships between MLO proteins were inferred using the maximum likelihood method based on the Whelan and Goldman model [53], using the MEGA6 software (http://www.megasoftware.net, accessed on 1 June 2023) [54]. The model with the lowest BIC (Bayesian Information Criterion) score was considered to describe the substitution pattern. The bootstrap consensus tree, obtained from 100 replicates, was taken to represent the MLO family phylogenetic history [55].

### 5.4. De Novo Prediction of MLO-Encoding Genes Motifs

The Multiple EM for Motif Elicitation (MEME) (https://meme-suite.org/, accessed on 1 June 2023) algorithm was used to decompose in motifs (Appendix A) the MLO Pfam domain (PF03094) of protein sequence data set. The analysis was carried out using the default cut-off value for statistical confidence. Motif Alignment and Search Tool (MAST) (https://meme-suite.org/, accessed on 1 June 2023) was also used to confirm the presence of MEME-motifs previously identified (Appendix A), using default setting [56]. The MEME-motifs with less than 70% site coverage were eliminated from the motif structure of phylogenetic clades.

### 5.5. Evolution Rates at Codon Sites

Evolutionary forces acting on MLO homologs, in *Lamiaceae* plant family, were investigated by determining two parameters based on the number of non-synonymous and synonymous substitutions per non synonymous and synonymous site (dN and dS, respectively), indicated as ω (dN/dS). Tests were conducted to estimate the evolution of each codon: positive (ω > 1); neutral (ω = 1); and negative (ω < 1). All MLO-coding DNA sequences were aligned using ClustalW (v2.0) [52], and positions with less than 80% site coverage were eliminated from the analysis. To clearly depict the proportion of sites under selection, an evolutionary fingerprint analysis was carried out using the implemented SLAC algorithm in the Datamonkey server at the default value [34]. Pressure selection analyses were performed on the same sequence subset of phylogenetic analysis. The average nucleotide diversity (π) was computed using the Tajima’s Test of Neutrality method implemented in the MEGA v6 software [54].

## Figures and Tables

**Figure 1 ijms-24-13627-f001:**
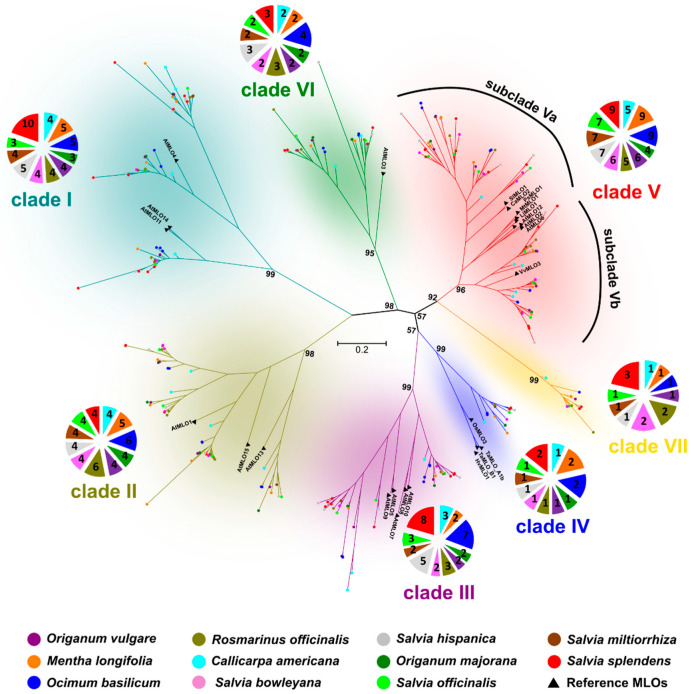
Maximum likelihood analysis of *Lamiaceae* MLO proteins. The tree includes 247 *Lamiaceae* MLO homologs and 25 reference MLO proteins already characterized in other species. Clades were enumerated with the Roman numerals from I to VII on the base of position of Arabidopsis and monocot MLO genes. The tree was drawn to scale, with branch lengths proportional to the number of substitutions per site. Bootstrap values of collapsed clades are indicated above the branches. The taxa to which the protein sequences belong are indicated by colored spots. Pie charts illustrate the number of MLO genes belonging to a given clade in each taxonomic group.

**Figure 2 ijms-24-13627-f002:**
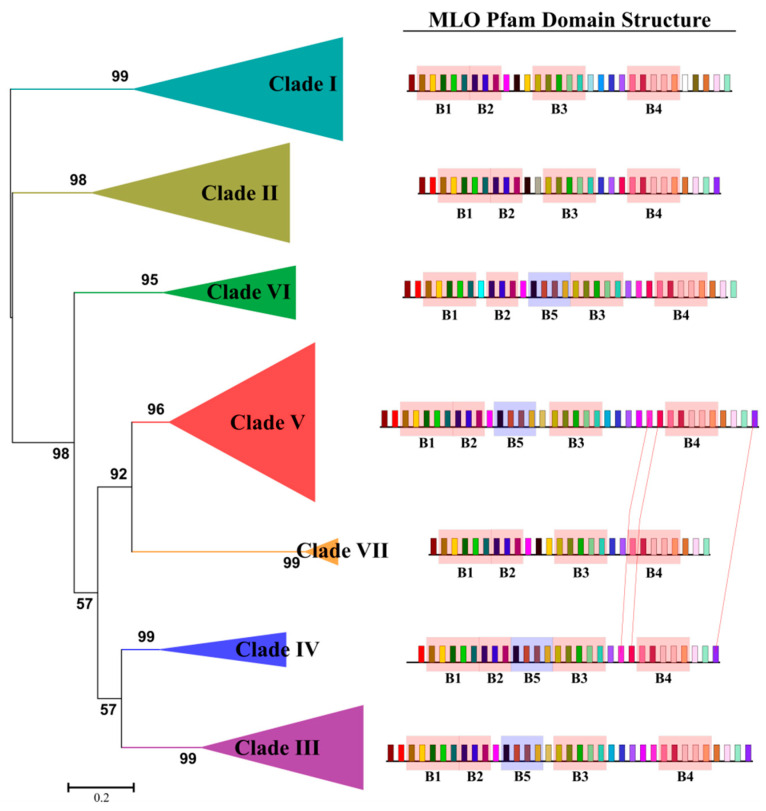
Identification and distributions of clade-specific amino acid motifs in MLOs. The six phylogenetic clades were collapsed and marked with different colors. For each clade the sequence logos of highly conserved motifs were showed. Black lines and colored boxes denote MLO domain sequences and motif positions, respectively. Red and blue rectangles delimit the conserved motif patterns between clades. Red lines connect motifs conserved between clades IV and V.

**Figure 3 ijms-24-13627-f003:**
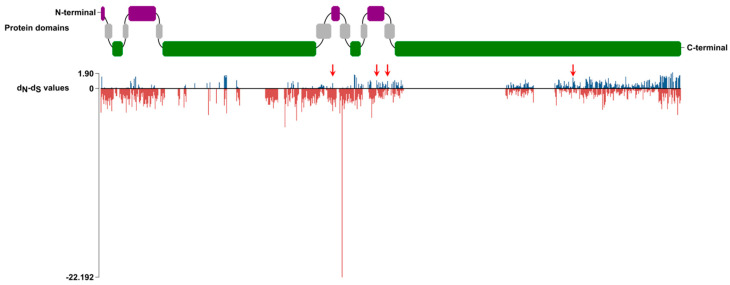
Natural selection acting on *Lamiaceae* clade V MLO homologs. Direction and magnitude of positive and negative dN–dS values are showed with blue and red lines, respectively. Extracellular, transmembrane and intracellular MLO protein regions are indicated with the violet, grey and green boxes, respectively. The red arrows indicate codons characterized by a significant *p*-value (*p* < 0.1) for positive selection (in position 955, 1135, 1180 and 1944 of the alignment).

**Table 1 ijms-24-13627-t001:** Number of MLO genes identified in eleven *Lamiaceae* genomes. Genome size and predicted gene sets were reported for each plant species.

Species	Common Name	Genome Size (Mb)	Predicted Gene Set	Number of MLO Loci	Reference Genome
*Callicarpa americana*	American beautyberry	506	32,164	23	Hamilton et al., 2020 [26]
*Mentha longifolia*	Silver mint	470	52,520 *	33	Vining et al., 2016 [23]
*Ocimum basilicum*	Basil	2068	78,990	45	Bornowski et al., 2020 [25]
*Origanum majorana*	Marjoram	761	33,929	21	
*Origanum vulgare*	Oregano	630	32,623	20	
*Rosmarinus officinalis*	Rosemary	1014	51,389	29	
*Salvia bowleyana*	Nan-dan-shen	462	44,044	22	Zheng et al., 2021 [28]
*Salvia hispanica*	Chia	348	31,069	28	Wang et al., 2022 [22]
*Salvia miltiorrhiza*	Dan-shen	595	32,483	21	Song et al., 2020 [24]
*Salvia officinalis*	Sage	480	50,957 *	28	Li et al., 2022 [29]
*Salvia splendens*	Scarlet sage	808	56,267	54	Jia et al., 2021 [27]
Total				324	

* Number of gene models predicted in this study by Augustus on *Mentha longifolia* [30] and *Salvia officinalis* [29] genome assemblies.

**Table 2 ijms-24-13627-t002:** Structural properties of MLO gene models identified in eleven genomes. Gene, protein and domain lengths, and number of exons were reported for each *Lamiaceae* species. The average number refers to the arithmetic mean between the members of each MLO gene family.

Species	Genomic Length (bp)	Number of Exons for Locus	Protein Length (aa)	Domain Length (aa) *
*Callicarpa americana*	5084	12	458	387
*Mentha longifolia*	4763	12	507	334
*Ocimum basilicum*	4101	11	407	335
*Origanum majorana*	3960	11	433	348
*Origanum vulgare*	4215	13	482	404
*Rosmarinus officinalis*	3591	11	412	341
*Salvia bowleyana*	5105	13	436	344
*Salvia hispanica*	3958	13	494	406
*Salvia miltiorrhiza*	5008	14	520	449
*Salvia officinalis*	4167	12	474	344
*Salvia splendens*	4082	11	454	343

* Length of MLO Pfam domain [31] predicted in this study by InterProScan [32].

**Table 3 ijms-24-13627-t003:** Estimation of evolutionary divergence (π) and non-synonymous to synonymous substitutions (ω) mean dissimilarity for each MLO gene family and phylogenetic clade.

Aligned Coding Sequences	MLO Loci (n.)	Average Identity (%)	Π *	ω	PSCs	NSCs
*C. americana* MLO family	23	39.2	0.364	0.223	0	247
*M. longifolia* MLO family	33	25.7	0.428	0.349	0	192
*O. basilicum* MLO family	45	29.9	0.398	0.250	0	243
*O. majorana* MLO family	21	33.2	0.374	0.251	2	292
*O. vulgare* MLO family	20	37.9	0.41	0.212	0	250
*R. officinalis* MLO family	29	34.3	0.369	0.249	2	180
*S. bowleyana* MLO family	22	36.7	0.369	0.259	0	182
*S. hispanica* MLO family	28	35.9	0.407	0.257	1	253
*S. miltiorrhiza* MLO family	21	45.1	0.431	0.221	2	264
*S. officinalis* MLO family	28	33.7	0.409	0.256	0	264
*S. splendens* MLO family	54	29.9	0.342	0.311	4	267
*Lamiaceae* MLO clade I	51	48.9	0.314	0.307	0	279
*Lamiaceae* MLO clade II	49	55.6	0.307	0.267	0	311
*Lamiaceae* MLO clade III	39	50.8	0.268	0.291	0	255
*Lamiaceae* MLO clade IV	14	68.3	0.146	0.236	0	96
*Lamiaceae* MLO clade V	73	44.2	0.254	0.375	4	371
*Lamiaceae* MLO clade VI	27	53.6	0.252	0.369	0	178
*Lamiaceae* MLO clade VII	13	73	0.115	0.216	0	65

* All positions with less than 80% site coverage were eliminated. The values of ω (=dN/dS) reported in the table were calculated using SLAC method and the number of positive (PSCs: positively selected codons) and negative (NSCs: negatively selected codons) codons statistically significant (*p* < 0.1) in MLO protein-coding genes, and are significant.

## Data Availability

The data generated and presented in this study are available in the Appendix A.

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
