# Peer review of "The First Genome-Wide Mildew Locus O Genes Characterization in the Lamiaceae Plant Family"

_ijms, 2023, doi:10.3390/ijms241713627_

Round 1
Reviewer 1 Report
The article "The first genome-wide MLO genes characterization in the Lamiaceae plant family" is well written and the subject interesting and innovative. However, I have noticed the lack of strong conclusions and speculations relating to the themes reported in the introduction: PM-Lamiaceae interaction, PM resistance, possible loci useful in breeding programs.
ABSTRACT:
Line 11: eliminate "of"
Lines 21-23: these issues are hardly addressed/discussed in the discussion section at all (see comments on the Discussions section)
INTRODUCTION:
Lines 30-32: eliminate “Indeed,…season[2]”, this sentence is not relevant and makes the paragraph more confusing
Lines 62-64: please write species names in italics
RESULTS:
Line 72: here and along all the text, should the name “Lamiaceae” be written in italics?
Line 92: please explain the acronym Pfam
Lines 107-109 please write species names in italics, check along all the text and be consistent
Line 148: correct “Figura 1”
Fig. 1 the quality of the figure is very low, please provide a better quality image. Furthermore, all writing and symbols (dots and triangles) should be larger, it is almost impossible to distinguish them. Also the number inside the pie charts are almost illegible.
Lines 215-216 please correct “in the in V and VII clades”
Lines 226-229: the caption of the Table 3 is not clear, please rewrite it
Figure 3: what are the blue and red dots mentioned in the description of the figure? Maybe you mean lines
DISCUSSION:
Lines 241-244: the sentence is not clear, please rewrite
Line 261: the sentence is not clear, please rewrite
Lines 270-282: these are probably the most important and interesting conclusions of your work, please extend more the explication and mention some of the results that validate these conclusions (figures, tables). This part of the discussion is very difficult to follow.
I. What are the "potential targets" for future breeding activities?
II: Is the "prediction" about the highly conserved amino acid motifs supported by previous literature? Please extend this part.
Lines 292-303: this part is very interesting, should be possible extend it more? Add more references and formulate more hypothesis about the PM-Lamiaceae relation
In general, the discussion section is very poor, the paper reports a large section of results and their discussion is very brief. Moreover, no speculations/ hypotheses relating to the objects of the work (PM-Lamiaceae relation, PM resistance, possible loci useful in breeding programs) are formulated on the basis of the results found
MATERIALS AND METHODS:
Line 321: please specify the other species (Hordeum, Oryza, Pisum…)
In general the manuscript is well edited and understandable.
Author Response
Dear Editor,
enclosed please find the revised version of manuscript ijms-1985667 "The first genome-wide MLO genes characterization in the Lamiaceae plant family" by Andolfo Giuseppe and Ercolano Maria Raffaella. During revision, we have done our best to address the valuable comments of the reviewers. Below, we provide a point-to-point reply to each comment. Changes with respect to the previous version are highlighted in yellow. Overall, we believe that the revised manuscript fully addresses the comments and we are looking forward to seeing our work published in the International Journal of Molecular Science.
Best regards,
Maria Raffaella Ercolano on behalf of two co-authors
Reviewer#1
The article "The first genome-wide MLO genes characterization in the Lamiaceae plant family" is well written and the subject interesting and innovative. However, I have noticed the lack of strong conclusions and speculations relating to the themes reported in the introduction: PM-Lamiaceae interaction, PM resistance, possible loci useful in breeding programs.
- We appreciate the time Reviewer#1 has spent in reading and revising this manuscript and we believe that their suggestions helped to improve our manuscript. All changes in the revised version are highlighted in the text. Please, find our responses to each of your suggestions/comments below.
ABSTRACT:
Line 11: eliminate "of"
- We correct it.
Lines 21-23: these issues are hardly addressed/discussed in the discussion section at all (see comments on the Discussions section)
- We agree with this point and have substantially revised the entire manuscript, especially the discussion section.
INTRODUCTION:
Lines 30-32: eliminate “Indeed,…season[2]”, this sentence is not relevant and makes the paragraph more confusing
Lines 62-64: please write species names in italics
- We correct it.
RESULTS:
Line 72: here and along all the text, should the name “Lamiaceae” be written in italics?
- We correct it.
Line 92: please explain the acronym Pfam
- We correct it.
Lines 107-109 please write species names in italics, check along all the text and be consistent
- We correct it.
Line 148: correct “Figura 1”
- We correct it.
Fig. 1 the quality of the figure is very low, please provide a better quality image. Furthermore, all writing and symbols (dots and triangles) should be larger, it is almost impossible to distinguish them. Also the number inside the pie charts are almost illegible.
- We have modified the Figure 1 to keep consistent according to suggestions. The size of all writing and symbols has been substantially incremented. Unfortunately, the colored dots cannot be enlarged, they would end up overlapping. However, a supplementary high-quality figure (Suppl. Figure 1) of the phylogenetic tree was provided, in which the gene IDs were also indicated.
Lines 215-216 please correct “in the in V and VII clades”
- We correct it.
Lines 226-229: the caption of the Table 3 is not clear, please rewrite it
- We have rewritten the caption of the Table 3.
Figure 3: what are the blue and red dots mentioned in the description of the figure? Maybe you mean lines
- We correct it.
DISCUSSION:
Lines 241-244: the sentence is not clear, please rewrite
- We rewrote the sentence to make it more readable and understandable.
Line 261: the sentence is not clear, please rewrite
- The sentence has been rephrased in the revised version of the manuscript.
Lines 270-282: these are probably the most important and interesting conclusions of your work, please extend more the explication and mention some of the results that validate these conclusions (figures, tables). This part of the discussion is very difficult to follow.
- What are the "potential targets" for future breeding activities?
- We thank the Reviewer#1 for pointing out this inconsistency, and we extended the explication and mention our results that validate these conclusions.
II: Is the "prediction" about the highly conserved amino acid motifs supported by previous literature? Please extend this part.
- For the first time, we have explored the evolutionary routes emerging in the phylogenetic tree of MLO family. We improved the discussion reporting a speculation on functional activity and motif structure variability.
Lines 292-303: this part is very interesting, should be possible extend it more? Add more references and formulate more hypothesis about the PM-Lamiaceae relation.
- We emphasized the importance of the achieved results (MLO candidates for PM resistance) by discussing them in more detail.
In general, the discussion section is very poor, the paper reports a large section of results and their discussion is very brief. Moreover, no speculations/ hypotheses relating to the objects of the work (PM-Lamiaceae relation, PM resistance, possible loci useful in breeding programs) are formulated on the basis of the results found
- We acknowledge the reviewer for their useful comments and suggestions, that, in our opinion, helped to improve the overall quality of the discussion section. We have gone through the concerns and tried to clarify murky points.
MATERIALS AND METHODS:
Line 321: please specify the other species (Hordeum, Oryza, Pisum…)
- We have spelled out the species reported in Supplementary Table 4.
Reviewer 2 Report
In the present study, authors characterized the mildew locus O (MLO) gene family in 11 lamiaceae species and sorted out the set of MLO gene families, and the associated regions for varying selection in susceptibility factors. The results are useful to find the potential candidate genes for breeding powdery mildew resistance. I recommend the manuscript with following suggestions.
Add the study background clearly and improve the introduction. Also, indicate the main points undertaken in the aims and objective section before the material and method section.
Authors should select several candidate genes and study the response/expression pattern by qRT-PCR analysis. If authors present the subcellular localization results of major genes, it will be more useful.
The abbreviation used must be explained on their first appearance, or provide a separate list of abbreviations.
Language need to be improved, particularly results and discussion section.
Language need to be improved, particularly results and discussion section.
Author Response
Dear Editor,
enclosed please find the revised version of manuscript ijms-1985667 "The first genome-wide MLO genes characterization in the Lamiaceae plant family" by Andolfo Giuseppe and Ercolano Maria Raffaella. During revision, we have done our best to address the valuable comments of the reviewers. Below, we provide a point-to-point reply to each comment. Changes with respect to the previous version are highlighted in yellow. Overall, we believe that the revised manuscript fully addresses the comments and we are looking forward to seeing our work published in the International Journal of Molecular Science.
Best regards,
Maria Raffaella Ercolano on behalf of two co-authors

Round 2
Reviewer 2 Report
The paper can be accepted
Minor editing of the english language required
Author Response
The paper can be accepted, however, it will be more significant if the authors do qRT-PCR.
- We thank the academic Editor for this valuable suggestion. In this work, we provide a genome-wide characterization of 11 Lamiaceae MLO gene families, and study their diversity in an evolutionary key. As far as we know, this is the first study about MLO annotation in Lamiaceae plant family and the first work that simultaneously analyzes such a large number of genomes. The expression study finalized to PM-susceptible factors identification will be realized in future.